# CCKS: Cooperative CPU-GPU Scheduling for Fused Kernels on Coherent Architectures

## Abstract

Executing modern ML workloads as sequences of discrete GPU kernels leads to significant hardware underutilization because of kernel launch, data movement, and CPU-GPU synchronization overheads. Recent advancements in kernel fusion reduce small kernel launch overhead by consolidating many small kernels into a single, persistent kernel. However, existing fusion techniques delegate complex scheduling logic to the GPU itself—a task for which its architecture is ill-suited. This on-GPU scheduling creates critical inefficiencies, as its control-intensive, synchronization-heavy logic is fundamentally mismatched with the GPU's parallel microarchitecture, and leads to stalled threads during synchronization, and high-overhead collection of global state.

We propose CCKS(Cooperative Coherent Kernel Scheduler), a novel framework that leverages tightly integrated, cache-coherent CPU-GPU architectures such as the NVIDIA Grace Hopper Superchip for fused kernel scheduling. CCKS offloads the scheduling of fused kernels to the host CPU, treating it as a dedicated co-processor. We introduce a pluggable scheduling programming interface that allows developers to easily offload complex GPU kernel scheduling logic on the CPU. We develop multiple scheduling optimizations that efficiently overlap CPU-GPU communication time with GPU's computation to improve performance. Our evaluation shows that CCKS achieves up to 77% improvement over state-of-the-art kernel fusion frameworks on representative ML workloads.

## 1 Introduction

Enabling high-performance execution of deep neural networks on GPUs is critical for modern ML applications. To maximize the performance of large-scale models like LLMs (Dao et al., 2022) and Vision Transformers (Dosovitskiy et al., 2020), modern GPU programming has shifted from launching many small, discrete kernels to using *fused persistent kernels* (NVIDIA, 2025; Jia, 2025; Spector et al., 2024; Kamath et al., 2025; Nrusimha et al., 2025). By combining multiple computational steps (e.g., attention, matrix multiplication, and reductions) into a single, persistent kernel launch, this approach eliminates repeated kernel launch overhead and improves hardware utilization. This paradigm, however, requires a sophisticated *on-GPU scheduling runtime* to manage the complex task graph within the kernel, dispatching computational tasks to different Streaming Multiprocessors (SMs) based on task dependencies and runtime behaviors (Wu et al., 2025; Spector et al., 2024).

Currently, these scheduling runtimes are implemented directly on the GPU; however, this creates a fundamental architectural mismatch, as *GPUs are not designed to execute control-intensive scheduling tasks*. GPUs are massively parallel processors designed for data-parallel, SIMT (Single Instruction, Multiple Threads) execution. Scheduling, in contrast, is an inherently serial, control-flow-heavy task dominated by branching and state management. Executing this logic on a GPU leads to severe hardware inefficiency, where only a few threads in a GPU warp are active, resulting in abysmal hardware efficiency (Jia, 2025). Furthermore, using GPUs to perform scheduling wastes resources, as the SMs dedicated to this logic (which can consume up to 18.2% of total GPU SMs) cannot be used for meaningful LLM computation, thereby slowing down the overall workload (Mirage-project, 2025).

A natural solution for reducing GPU scheduling overhead is to offload this logic to a CPU, which is explicitly designed for complex, sequential control flow. Historically, this has been infeasible due to the high communication overhead of traditional CPU-GPU interconnects like PCIe, which incurs microseconds overheads coming from the CPU issuing kernel launches, managing explicit data transfers (e.g., cudaMemcpy), synchronizing with the GPU, and the physical interconnect's delays for data transfer (Lustig & Martonosi, 2013; Hwang et al., 2023) . Fortunately, a recent hardware trend toward tightly-integrated CPU-GPU systems, such as the NVIDIA Grace Hopper Superchip (Nvidia, 2023), presents a new opportunity . These architectures feature ultra-low latency, high-bandwidth, cache-coherent interconnects (MVLink Chip-to-Chip (NIVIDIA, 2025))that transform the CPU from a distant host into a tightly-coupled co-processor (Schieffer et al., 2024; Fusco et al., 2024; Choi et al., 2025). This enables frequent, fine-grained communication between a CPU-based scheduler and GPU-based workers with minimal overhead.

To improve fused kernel performance, as well as harness the new capability of tightly integrated CPU-GPU systems, we present CCKS (Cooperative Coherent Kernel Scheduler), a scheduling framework that makes high-performance, CPU-assisted GPU scheduling practical. CCKS provides a simple, pluggable API that allows persistent kernel developers to offload complex scheduling logic to the CPU with minimal code changes. Furthermore, CCKS introduces two key optimizations designed to improve performance by overlapping scheduling with GPU computation, specifically:

1. We propose CCKS, a fused kernel scheduling framework that, for the first time, enables high-performance scheduler offloading from a persistent GPU kernel to a tightly integrated CPU kernel ( §3).
2. We design a pluggable scheduling programming interface that allows developers to easily implement complex scheduling logic on the CPU, while the framework provides highly efficient CPU-GPU communication via shared memory queues over cache-coherent interconnects like NVLink Chip-to-Chip ( §3.1).
3. We develop multiple scheduling optimizations in CCKS. The first is *Speculative Enqueue and Batch Commit*, which reduces GPU-CPU communication overhead by having the CPU scheduler speculatively enqueue tasks to overlap CPU-GPU communication time with GPU's computation, thereby reducing scheduling delays and reduces bubbles for GPU's execution pipeline ( §3.2.1).
4. Another optimization we propose is *Scheduler Bypass*, which uses direct GPU-to-GPU communication to bypass the CPU scheduler, which entirely eliminating the CPU round-trip for tasks with static dependencies. This optimization is applicable when the task graph can be determined ahead of time without dynamic CPU runtime scheduling decisions ( §3.2.2).
5. Our experiments demonstrate the effectiveness of CCKS on modern AI workloads and compare it with popular fused kernel frameworks that use GPU scheduling (Wu et al., 2025; Kamath et al., 2025; Spector et al., 2024). We show that compared to state-of-the-art on-GPU schedulers, CCKS improves end-to-end performance by up to 77% and enabling flexible and more efficient scheduling policy on CPU ( §4).

## 2 MOTIVATION

### 2.1 FROM DISCRETE KERNELS TO DYNAMIC PERSISTENT KERNELS

**Limitations of Discrete Kernels and CUDA Graphs:** Traditional inference systems for LLMs execute a sequence of discrete GPU kernels to perform computation. However, this approach introduces significant performance bottlenecks because each kernel launch incurs setup and tear-down costs, consuming several microseconds per call (Jia, 2025; Spector et al., 2024).

To mitigate these overheads, software frameworks such as CUDA Graphs NVIDIA (2020) provide APIs that bundle a sequence of kernels and operations into a single executable graph, which can be launched with a single `cudaGraphLaunch` call. This approach effectively amortizes CPU-side dispatch costs and addresses launch overheads.

The challenge with this CUDA Graphs approach is that the graph must be constructed in full (including dependencies, kernel launches, and barriers) before the application is executed on the GPU for each input. As a result, CUDA Graphs cannot adapt to runtime dynamics (*e.g.,* different inputs, tail effects of kernel execution), which can lead to performance degradation (Durvasula et al., 2024; Spector et al., 2025).

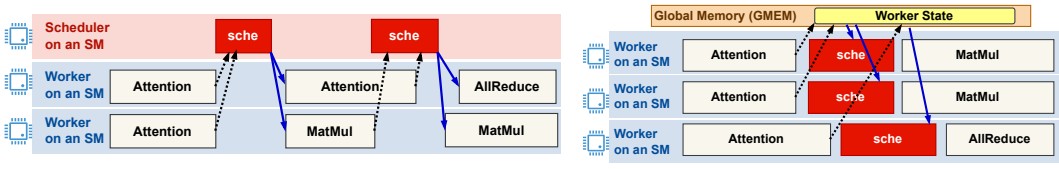

(a) Dedicate GPU Scheduler

(b) Coroutine GPU Scheduler

Figure 1: Kernel Fusion Scheduling.

**Towards Dynamic Persistent Kernels** To overcome the static nature of CUDA Graphs and better adapt to runtime dynamics, recent work proposes transforming LLM inference into a dynamic, persistent, fused kernel (NVIDIA, 2025; Jia, 2025; Spector et al., 2025). In this model, a grid of thread blocks is launched once and remains resident on the GPU's SMs for the duration across inference requests. A persistent thread-block scheduler dynamically assigns work, allowing the kernel to express custom, fine-grained data dependencies and make scheduling decisions at runtime. This approach eliminates kernel launch overhead and leverages a persistent scheduler to adapt to changing workloads.

Persistent kernels have been widely adopted in recent systems Jia (2025); Spector et al. (2024); Kamath et al. (2025); Nrusimha et al. (2025) and are supported in the NVIDIA CUTLASS library on the latest Blackwell GPUs (NVIDIA, 2025).

### 2.2 Persistent Kernel Scheduling and Its Overhead

The core innovation enabling kernel fusion is an *on-GPU scheduler* that dynamically executes a task graph entirely within the persistent kernel (Jia, 2025; Spector et al., 2025). This scheduler runtime allows for fine-grained control over task execution and scheduling on Streaming Multiprocessors (SMs) without host intervention. An on-GPU scheduler can dynamically determine the order of operations and dispatch tasks to available SMs, effectively hiding latency and preventing straggler blocks from stalling forward progress. There are multiple strategies for implementing this on-GPU runtime scheduler, as illustrated in Figure 1.

1. **Dedicated Scheduler** (Figure 1a): This model designates one or more SMs to act exclusively as schedulers (Jia, 2025). After worker SMs complete a task, they update their status in a shared memory region. The dedicated scheduler SMs monitor this region, determine which subsequent tasks have their dependencies satisfied, and dynamically dispatch them to available worker SMs. The SMs uses for scheduling cannot be used for LLM computation. For instance, the existing system requires 12 scheduler SMs to manage 96 worker SMs (Mirage-project, 2025), sacrificing over 12% of the GPU's computational power.

2. **Coroutine Scheduler** (Figure 1b): In this model, scheduling logic is embedded directly within the worker code on each SM (Kamath et al., 2025). The existing NVIDIA CUTLASS library also employs such a scheduling mode via Blackwell Cluster Launch Control (CLC) (NVIDIA, 2025). After a SM worker completes a computational task, the SM updates its status in global memory and then switches context to a *scheduler coroutine*. This scheduler coroutine inspects the global memory to learn the progress of other workers to determine the next task to execute on its own SM. In state-of-the-art coroutine scheduler implementations (Kamath et al., 2025), this scheduling phase accounts more than 11% of the total GPU cycles.

**Inefficient on-GPU scheduling.** Both on-GPU scheduling approaches introduce significant resource inefficiency and performance degradation. To quantify this, we profiled state-of-the-art fused kernels that use dedicated schedulers (Jia, 2025) or coroutine schedulers (Kamath et al., 2025). As shown in Figure 2, our analysis reveals that on-GPU scheduling causes hardware utilization to drop by 19% to 60%. This drop occurs for two main reasons: 1) the scheduler logic consumes GPU resources (either entire SMs or execution cycles), making them unavailable for model computation, and 2) as GPU tasks wait to be scheduled, bubbles are created in the execution pipeline, which directly translates to a slowdown of the primary workload. The significant inefficiency of on-GPU scheduling shows that there is substantial room for improving fused kernel performance.

*The fundamental reason for such resource inefficiency stems from a deep architectural mismatch.* GPUs are massively parallel processors designed for data-parallel, Single Instruction, Multiple

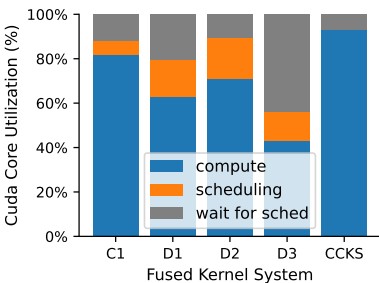

Figure 2: Overhead of scheduler in different fused kernel systems. C1 stands for co-routine scheduler system (Kamath et al., 2025) and D1,D2,D3 stand for different dedicated scheduler systems (Mirage-project, 2025).

| Lat Breakdown | PCIe Gen5 | NVLink C2C |
|---|---|---|
| SW Overhead | 3.8 μs | 1.0 μs |
| Phy Link | 1.0 μs | 800 ns |
| Sync Cost | 11 μs | 700 μs |
| **Total CPU-GPU Lat** | **15.8 μs** | **2.5 μs** |
| **Aggregate Bandwidth** | **128 GB/s** | **900 GB/s** |

Figure 3: A Comparison of CPU-GPU Communication Latency and Bandwidth between PCIe and NVLink C2C.

Thread (SIMT) workloads. Scheduling, in contrast, is an inherently serial and control-flow-heavy task dominated by branching and state checks. When this logic is mapped to a GPU's SIMT architecture, it causes severe warp divergence. Only a single thread in a warp (a group of 32 threads) may be active while executing a specific control path, leaving the other 31 threads idle. This results in abysmal hardware utilization for the scheduling logic, as low as 3% (1/32), wasting the vast majority of the SM's potential throughput.

As shown in Figure 2, our CCKS system will overcome this on-GPU scheduling limitation and dramatically improve the GPU hardware utilization.

### 2.3 A CASE FOR KERNEL SCHEDULING OFFLOADING

A natural solution to reduce the overhead of on-GPU scheduling is to offload the scheduling task to external hardware, such as a CPU, which is better suited for control-flow-heavy logic and can free up GPU resources for LLM computation. The primary challenge with this approach, however, has been high communication overhead between the CPU and GPU. Interacting with a GPU over a traditional interconnect like PCIe incurs a high latency, spanning hundreds of nanoseconds to microseconds, with the overheads coming from the CPU issuing kernel launches, managing explicit data transfers (e.g., cudaMemcpy), synchronizing with the device, and the physical interconnect's delays for data transfer. This high communication overhead reduces the effectiveness of offloading scheduling to the CPU, as it would force the GPU to stall while waiting for scheduling decisions, creating significant idle periods that would negate the benefits of the offload.

Fortunately, recent advancements in CPU-GPU co-packaging are changing this landscape. Tightly-integrated architectures, such as the NVIDIA Grace Hopper Superchip, connect the CPU and GPU via ultra-high-speed, coherent chip-to-chip (C2C) interconnects like NVLink-C2C (NIVIDIA, 2025; Huynh et al., 2025). In Figure 3 we measure and compares the communication latency of these integrated systems to that of traditional systems using PCIe. As the breakdown shows, coherent interconnects offer an order-of-magnitude lower latency and vastly higher bandwidth. Crucially, they also support hardware cache coherency, allowing the CPU and GPU to access each other's memory through simple, cacheable load/store instructions and to synchronize using low-overhead atomic operations over shared memory (without making expensive *cudaMemcpy( )* and *cudaStreamSynchronize*). This tight integration transforms the CPU into a viable co-processor for the GPU, paving the way for offloading fine-grained scheduling tasks without incurring huge data-movement overhead.

In this paper, we propose a method that utilizes the tightly integrated CPU to offload scheduling for fused kernel. This enables the benefits of chip-to-chip coherent interconnects and provides ultra-low overhead kernel scheduling.

### 3 CCKS DESIGN

In this section, we introduce CCKS (Cooperative Coherent Kernel Scheduler), a kernel scheduling framework designed to leverage tightly-integrated, cache-coherent CPU-GPU architectures, such as

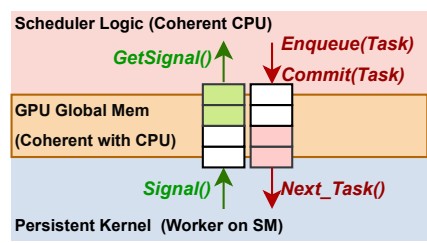

(a) CCKS Interface.

```
while(true) {
    // Get a Task from Scheduler
    task = Next_Task();
    if (task == TASK_ATTENTION) {
        _kern_attention(...);
    } else if(task = TASK_REDUCE){
        _kern_allreduce(...);
    }
    // Update state to scheduler
    Signal(TASK_END);
}
```

(b) Invoke CCKS interface from GPU kernel.

Figure 4: CCKS overview, shows the interface (a) and the example kernel program uses CCKS (b).

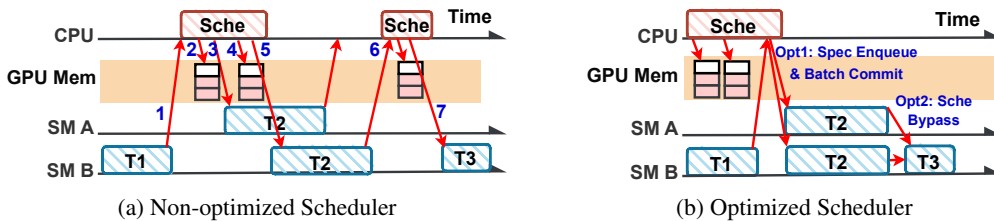

(a) Non-optimized Scheduler

(b) Optimized Scheduler

Figure 5: Scheduling Optimizations.

the NVIDIA Grace Hopper Superchip. The design of CCKS is twofold. First, it provides a simple, pluggable API that allows persistent kernel developers to offload complex scheduling logic to the CPU with minimal code changes. Second, it incorporates a set of runtime optimizations (*e.g.,* pre-enqueue, scheduling-bypass, and just-in-time scheduling) that remove the CPU scheduler from the critical data path when possible, minimizing the time GPU workers spend stalled while waiting for their next task to be scheduled.

### 3.1 PLUGABLE SCHEDULING API

Figure 4a shows the interface of CCKS. The communication interface between the CPU scheduler and GPU workers is built upon a set of shared, asynchronous queue pairs. One is *signal queue*, which is used for GPU-to-CPU signal updates, and the other is *task queue*, which is used for CPU-to-GPU task assignments. Each GPU worker SM assigned with a dedicated queue pair to communicate with the CPU scheduler. These queues reside in GPU global memory (e.g., HBM), which the CPU can access directly through regular, cacheable load/store instructions, thanks to the hardware cache coherency of the interconnect.

CCKS provides a set of primitives for the GPU kernel and the on-CPU scheduler to interact with the queue pairs. On the GPU, a worker kernel calls *Next_Task()* to fetch its next task from the *task queue*. The task includes the operation type (e.g., attention, matrix multiplication), as well as input parameters for the operation (e.g., control flags, tensor offsets, batch ID). The GPU worker also uses *Signal()* to notify the CPU about task completion or progress through the *signal queue*. Figure 4b shows an example of a GPU kernel program that leverages this interface.

On the CPU, the scheduler runtime calls *GetSignal()* to retrieve the status of GPU workers from the *signal queue*. Based on the workers' progress, the scheduler makes its decisions, copies new tasks into the task queue using *enqueue(Task)*, and finally calls *Commit()*. The *Commit()* operation advances the queue's tail pointer, making the newly enqueued tasks visible to the GPU workers.

### 3.2 SCHEDULING OPTIMIZATIONS

While keeping the programming interface simple, under the hood CCKS is flexible about *how the interface is executed*. To maximize performance, CCKS incorporates two optimizations designed to overlap scheduling with GPU computation: **Pre-enqueue & Batch Commit** and **Scheduler Bypass**.

Figure 5a illustrates the default CCKS scheduling timeline without these optimizations. This example involves three tasks (T1, T2, and T3), where each subsequent task executes only after all instances of the preceding task have completed. These tasks execute on the GPU's SM_A and

```
1   import CCKS as ccks
2   class MyScheduler(ccks.Scheduler):
3       # T2's scheduler which is non-static
4       # Runs on CPU, will not enable sche_bypass opt
5       userInput = input_queue() # read from the network req
6       @ccks.signal('T1_TASK_END') # scheduler triggered by T1_TASK_END
7       def sche_t2_non_static(_ : ccks.Signal):
8           c = userInput.fetch_next() # non-state access lead to non-static operation
9           if c.type == 'RUN_T2':
10              ccks.enqueue(ccks.Task('T2'),queue['SM_A'])
11              ccks.enqueue(ccks.Task('T2'),queue['SM_B'])
12              ccks.batch_commit(queue['SM_A', 'SM_B']) # batch commit to SMs
13
14      # T3's scheduler which is static
15      # Runs on GPU, will enable sche_bypass opt
16      @ccks.signal('T2_TASK_END') # scheduler triggered by T2_TASK_END
17      def sche_t3_finish(_ : ccks.Signal):
18          c = ccks.get_state('t2_counter')
19          c += 1
20          if c >= 2: # T2's two tasks finishes
21              ccks.enqueue(ccks.Task('T3'),queue['SM_B'])
22              ccks.commit(queue['SM_B'])
```

Figure 6: Example user code that uses CCKS to implement scheduling logic in Figure 5b

SM_B. The figure shows the steps of tasks scheduling: 1) T1 uses *Signal()* to notify the CPU about its completion. 2) To schedule two instances of T2 on SM_A and SM_B, the CPU first uses *enqueue()* to copy T2 from the CPU into SM_A's task queue in GPU memory. 3) The CPU then uses *Commit()* to make T2 visible to SM_A. 4) Similarly, the CPU enqueues T2 to SM_B's task queue and 5) commits it to SM_B. 6) SM_A and SM_B signal the CPU scheduler once T2 finishes. 7) Once the final instance of T2 finishes, the CPU scheduler enqueues T3 and commits it to SM_C.

This baseline can introduce significant inefficiency due to long scheduling delays from two sources:

- **Long Enqueue and Commit Time:** When the CPU scheduler needs to schedule tasks to multiple SMs, it must repeatedly perform *enqueue* and *commit* operations for each task. Each of these operations involves a distinct memory copy from CPU to GPU, and this serialization prolongs the time the CPU spends on scheduling, delaying task launches.
- **CPU Round Trip Latency:** The launch of a successor task is delayed by a full CPU-GPU round-trip because every GPU task must wait for the CPU scheduler to make its decision. For example, in Figure 5a, T3 cannot be scheduled until the CPU processes the completion of T2, creating an execution bubble on the GPU.

To overcome these overheads, CCKS employs following two optimizations (shown in Figure 5b).

### 3.2.1 SPECULATIVE ENQUEUE AND BATCH COMMIT

The first optimization targets the overhead of sequential host operations by decoupling task preparation from task commit.

- **Speculative enqueue:** The CPU scheduler speculatively prepares task data structures and copies them into GPU memory queues *in advance*, even when prior GPU tasks are still running. As shown in Figure 5b, the tasks for T2 are pre-enqueued before T1 finishes, thereby overlapping the CPU-GPU communication with T1's computation.
- **Batch Commit:** Once T1 finishes, the scheduler uses a single, efficient *Batch Commit* operation. This is done by using vectorized instructions to update multiple task queue pointers with a single write, making all pre-enqueued tasks visible to their respective SMs simultaneously.

This speculative pre-enqueueing is robust; if the scheduling decision changes after T1 finishes, the framework allows the user to re-enqueue the correct task by overwriting the previously staged entry.

Furthermore, CCKS allows the user to choose whether to enable batch commit, trading off latency and throughput. For latency-sensitive workloads, batch commit can be disable to ensure each kernel are launched in a timely fashion to reduce latencies.

### 3.2.2 SCHEDULER BYPASS

Even with pre-enqueue and commit batching, scheduling often requires a CPU-GPU round trip. The *scheduler bypass* optimization eliminates this latency for tasks with static dependencies. This is applicable when the task graph can be determined ahead of time without dynamic runtime decisions.

As shown in Figure 5b, the dependency between T2 and T3 is static. The bypass mechanism allows the T2 tasks to directly signal and activate T3 on SM_C, completely avoiding the host CPU. This direct on-GPU signaling minimizes the launch latency for T3.

However, the scheduler bypass can only be employed in cases where scheduling dependencies between tasks are static. For scheduling dependencies that rely on non-static inputs (e.g., user or network input, output of a previous layer), workers still need to signal the CPU scheduler to make runtime decisions. The CCKS framework automates this process through a compiler-driven approach that involves two main stages:

1. **Static Scheduler Classification:** First, CCKS performs a static analysis on user-implemented scheduler functions to classify them as either *static* or *dynamic*. A function is classified as static if its scheduling decisions depend only on constants and internal scheduler state that can be manipulated with simple arithmetic (e.g., the T3 scheduler in Figure 6's line 17 is static). On the other hand, a function is classified as dynamic if its logic depends on external runtime inputs, such as in Mixture of Experts (MoE) models where the routing of a token determines the next computational task (e.g., the T2 scheduler in Figure 6's line 7 is dynamic). Only functions classified as static are eligible for the bypass optimization.
2. **Kernel Transformation:** For functions identified as static, the compiler rewrites the GPU kernels. The 'Signal()' call in the launcher task is transformed to execute the simple scheduling logic directly on its SM. The receiver task is prepended with a micro-kernel that polls the on-GPU state until the commit condition is met before proceeding.

## 4 IMPLEMENTATION AND EVALUATION

We implemented CCKS using 1.7K lines of CUDA C++ for the kernel and 2.3K lines of Python for the scheduling interface. Our implementation targets coherently connected CPU-GPU systems such as Grace-Hopper Superchips, and leverages cache-coherency interconnect to optimize communication. For example, we reduce the required number of atomic operations and memory barriers by leveraging the hardware's consistency guarantees and our batching optimizations (§6). This optimized implementation achieves performance near the physical limit of the interconnect, allowing the CPU scheduler to keep pace with the GPU's execution speed.

We integrated CCKS into three state-of-the-art fused-kernel systems: Pod-Attentio (Kamath et al., 2025), Mirage (Mirage-project, 2025), and ThunderMLA (Spector et al., 2025). Integration details and programming efforts are described in §4.1. Our testbed is an NVIDIA Grace Hopper GH200 system (Nvidia, 2023), which features an H100 GPU with 96 GB of HBM3 memory connected to a 72-core Grace ARM CPU via a 900 GBps cache-coherent NVLink-C2C interconnect.

| System | Scheduler Type | LoC (Sched) | LoC (Kern) |
|---|---|---|---|
| PodAttention | Non-Persistent, Co-Routine | 23 | 62 |
| Mirage | Persistent, Dedicated | 341 | 212 |
| ThunderMLA | Persistent, Offline | 242 | 64 |

Table 1: Example Integrated Fused-Kernel Implementations

### 4.1 INTEGRATION AND PROGRAMMING OVERHEAD

We show that the programming efforts of using the CCKS library to offload the scheduling of fused kernels to the host CPU. We adopt CCKS for three state-of-the-art systems, each representing a distinct fuse-kernel scheduling paradigm. As summarized in Table 1, it required only ≈200 Lines of Code (LoC) for each system to use the CCKS library to achieve scheduling offloads. The LoC can be split into two parts:

- **Scheduler LoC:** The implementation of the scheduling algorithm in the CCKS scheduler.

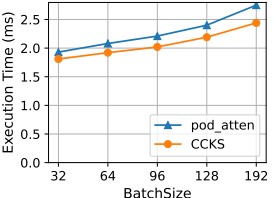
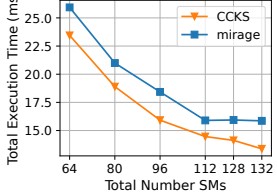
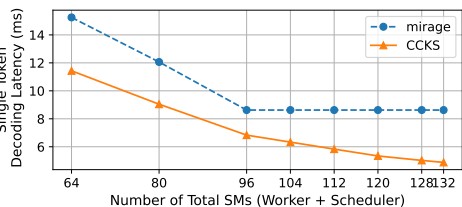

Figure 7: PodAttention Attention Kernel Performance Under Different Batchsize

Figure 8: Mirage Attention Kernel Performance Under Different Number of SMs

Figure 9: Mirage Results with latency-centric QWen3-8B Inference

- **Kernel LoC:** The modifications on the GPU SM workers to interface with the CCKS scheduler.

Both these LoCs remain low because the CCKS high-level interface abstracts away the complexity. Furthermore, the optimizations described in §3.2 (*e.g.,*, Speculative Enqueue, Batch Commit, and Scheduler Bypass) are built into the CCKS library. Thus, once using the CCKS API, these systems automatically inherit these optimizations without extra code changes. Below we describe more integration details:

**Integration with Pod-attention:** Pod-Attention utilizes a per-SM coroutine scheduler that dynamically selects between prefill and decode phases based on local execution ratios. We integrated CCKS by replacing this per-SM local coroutine scheduler with a global CPU-offloaded scheduler. Specifically, the native routine (in which each SM inspects a shared variable to determine the next task) was substituted with a single *ccks.Next_Task()* call within the main worker loop. This minimal modification enables Pod-Attention to operate in a fully persistent mode, effectively eliminating its primary performance bottleneck.

**Integration with Mirage:** Mirage originally uses a complex on-GPU dedicated scheduler, statically inferring a task graph and reserving dedicated SMs to manage execution using on-GPU counters and intricate worker-scheduler protocols. With CCKS, we removed Mirage's dedicated scheduler SMs entirely. The worker kernels were modified to simply call ccks.Signal() upon completion. The original scheduling logic is implemented by a CCKS scheduler loading Mirage-generated graphs.

**Integration with ThunderMLA:** ThunderMLA based on the ThunderKitten framework, is a fused kernel for MLA attention. ThunderMLA originally replies on a offline scheduler that uses performance modeling to generate a static execution plan, which is then executed by the GPU. CCKS seamlessly integrates with ThunderMLA's message format and brings real-time scheduling capabilities that outperform its static offline scheduler by leveraging live execution information.

### 4.2 END-TO-END RESULTS

In this section, we compare the end-to-end performance of CCKS against state-of-the-art fused kernel frameworks and aim to answer the following questions: 1) How does offloading scheduling to the CPU affect the performance of fused kernel (§4.2.1)? 2) What are the impacts of our proposed design decisions and optimizations(§4.2.3)? 3) What additional benefits does a CPU-based scheduler provide over GPU scheduler(§4.3)?

#### 4.2.1 CPU OFFLOADING PERFORMANCE GAIN

A primary benefit of CCKS is that it frees all of the GPU's SMs to be used for computation, unlike previous on-GPU schedulers that must reserve some of these resources for the scheduling logic. To demonstrate this, we enable CCKS for two systems, Pod-Attention and Mirage, implement the CPU scheduler to replicate these systems' native scheduling algorithms, and compare them with the original Pod-Attention and Mirage's GPU scheduling performance.

Figure 7 shows the performance comparison between CCKS and Pod-Attention with 8K context length input for the attention in the LLama-3-8B model (Dubey et al., 2024). Figure 8 shows the comparison with Mirage with the attention operator with 8K context length for the Qwen-8B model. For PodAttention, CCKS achieves 15% end-to-end execution time reduction when batch size equals 192. For Mirage, CCKS achieves 20% improvement. The key reason for these performance gains

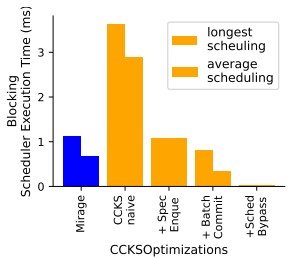

Figure 10: Breakdown of Scheduler Time Saving.

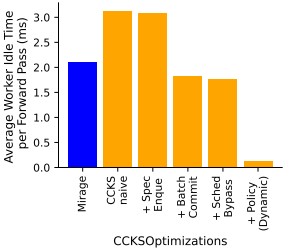

Figure 11: Breakdown of Worker Idle Time.

| Scheduler | B4 | B64 | Sched |
|---|---|---|---|
| Backward | 41.4 $\mu$s | 53.1 $\mu$s | 2.4 ms |
| CCKS-Static | 47.3 $\mu$s | 72 $\mu$s | 2.2 $\mu$s |
| CCKS-Dynamic | 44.3 $\mu$s | 52 $\mu$s | 23 $\mu$s |
| CCKS-Aware | 39.2 $\mu$s | 53.8 $\mu$s | 33 $\mu$s |

Figure 12: Single MLA kernel finish time compared with FlashMLA under different CCKS Scheduling Polices, with different **B**atch size. The Input and Execution **Aware** Policy Get Best Performance.

is that by offloading scheduling, the GPU resources previously consumed by the on-GPU scheduler are now available for the primary workload, leading to higher overall throughput.

We then measure how CCKS's performance scales with a varying number of SM workers. As shown in Figure 8, when we change the number of persistent kernel launched, the performance of CCKS continues to improve until all available SMs on the H100 GPU are utilized. In contrast, the performance of existing fused kernels, such as Mirage, stops improving beyond 112 workers. This is because Mirage must partition its SMs between workers and schedulers; the more workers it has, the more schedulers are needed, and more synchronization traffic must be exchanged between them. This result shows that CCKS's scheduling can scale with an increasing number of SM workers without introducing bottlenecks, allowing performance to improve as more resources are added.

Figure 9 further compares the performance of Mirage and CCKS on an end-to-end Qwen-8B inference workload (Yang et al., 2025), using a sequence length of 2048 tokens and a batch size of 16. The results show that CCKS provides up to a 77% performance improvement with 132 workers. Similar to the previous result, Mirage's performance stops increasing after allocating 96 SMs, while CCKS's performance continues to scale. This turning point comes earlier because the full inference job have more intra-task dependency, and triggers more intricate scheduling behavior, where CCKS and its optimizations collaboratively improve performance and avoid the bottleneck.

### 4.2.2 PERFORMANCE BREAKDOWN AND DEEP DIVE

Next, we analyze how CCKS optimizations improve CCKS's performance.

First, CCKS's optimizations accelerate its datapath. By moving operations out of the critical scheduling path, CCKS overcomes the higher physical latency of communicating from the CPU. Figure 10 compares the maximum and average scheduling latency of CCKS's decisions against Mirage's on-GPU scheduler. In the Qwen benchmark (Yang et al., 2025), the most complex scheduling job involves dispatching 20K tasks to 132 SMs. A naive CCKS implementation that simply mimics the on-GPU scheduler would incur 2.6x higher latency due to the CPU's slower access to GPU memory. However, our key optimizations greatly reduce this overhead. Since the scheduling pattern is fixed, speculative enqueuing decreases the latency by 2.7x. Batch-committing tasks (on average, 200 tasks per queue in this workload) provides an additional 1.1x improvement.

Furthermore, CCKS automatically identifies an optimization opportunity not exploited in the original Mirage implementation: the process of triggering the forward pass for next token is static and can be eliminated by our scheduler bypass optimization. This saves CCKS two round-trip to the scheduler, a reduction of 1.1 ms per token.

As shown in Figure 11, our optimizations yield a total scheduling time reduction of 70%. Beyond these datapath improvements, the flexibility of the CPU-based scheduler also helps eliminate worker idle time. Unlike static policies that must finish all tasks from one request before starting another, a dynamic CPU scheduler can use tasks from different requests to fill idle SMs. This dynamic policy boosts end-to-end performance by up to 78% without increasing latency.

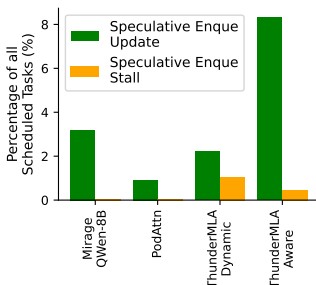

Figure 13: Percentage of Speculative Enqueue in Different Scheduling Tasks. *Orange bars shows SM's stall-and-wait caused by wrong speculation.*

| Instruction Sequence | CPU:H100 PCIe | CPU:H100 GraceHopper |
|---|---|---|
| Enque (16) | 32.4 $\mu$s | **4.2 $\mu$s** |
| Commit | 22.4 $\mu$s | **3.1 $\mu$s** |
| Update + Commit | 82.6 $\mu$s | **3.1 $\mu$s** |
| GetSignal + Update + Commit | 224.2 $\mu$s | **3.6 $\mu$s** |
| (End-to-End) Mirage Single Token Decoding | 57.24 ms | **4.882 ms** |

Figure 14: Latency of CCKS operations sequences under different types of CPU-GPU interconnects.

### 4.2.3 PERFORMANCE BREAKDOWN AND DEEP DIVE

### 4.3 NEW POLICY DESIGN SPACE WITH CPU SCHEDULER

With CCKS, fused kernel systems can unlock more flexible scheduling policies. Figure 12 shows CCKS's performance compared with FlashMLA under different scheduling policies. The result shows that CPUs are better suited for handling complex control flow and communication, and they have access to a wider variety of system information. For example, a CPU can deploy a performance model to predict the execution time of each kernel and create corresponding schedules, a task that is difficult to implement efficiently on a GPU. Here, we demonstrate how CCKS enables scheduling policies (CCKS-Aware) that can be dynamically adjusted based on information from both the host system and the live state of the GPU, and beat a long-running offline policy.

### 4.4 MIS-PREDICTION IN SPECULATIVE ENQUEUE

Figure 13 quantifies the mispredictions incurred by the speculative enqueue mechanism. As shown, the misprediction rate ranges from 1% to 8%; however, less than 1% of these instances actually lead to GPU SM stalls. This discrepancy exists because misprediction recovery is often handled off the critical path. Consequently, stalls are rare and primarily driven by dynamic graph features, such as the end-of-sequence token in Mirage Qwen Inference. In the worst case, speculative enqueue mispredictions account for less than 1.2% runtime overhead, which is a negligible cost compared to the 46% performance improvement brought by Speculative Enqueue.

### 4.5 IMPACT OF COHERENT INTERCONNECTION

In Figure 14, we evaluate the scheduler's performance across different interconnect architectures. As a baseline, we implement CCKS on a cloud-based H100 PCIe system, where command queues reside in pinned host memory mapped via the $cudaHostGetDevicePointer$ API. The results demonstrate that CCKS on GraceHopper significantly benefits from its hardware coherence. For example, the $Update + Commit$ operations are 28× faster on GraceHopper compared to the H100 PCIe system. This performance gap exists because: 1) GraceHopper's NVLink C2C has much lower latencies compared to PCIe, 2) CCKS's speculative enqueue requires modifying memory and signaling visibility via pointer updates. On PCIe-based non-coherent systems, the interconnect lacks efficient support for these atomic operations, incurring extra CPU-GPU round-trips to fetch data.

## 5 CONCLUSION

To address the significant performance overhead of on-GPU schedulers for fused kernel, this paper introduced CCKS, a framework that offloads GPU kernel scheduling to tightly-coupled CPUs. Through optimizations, CCKS minimizes communication costs and frees all GPU resources for computation. Our evaluation shows that CCKS improves end-to-end inference performance by up to 77% and achieves linear scalability, overcoming the thrashing bottlenecks of on-GPU approaches. This work demonstrates that leveraging the CPU as a first-class co-processor unlocks substantial performance gains for next-generation AI systems.

ETHICS STATEMENT

This material is based upon work supported by funding and gifts from multiple institutions. Any opinions, findings, conclusions, or recommendations expressed in this material are those of the authors and do not necessarily reflect the views of these institutions. Overall, the authors affirm that this work does not involve any ethical violations.

REPRODUCIBILITY STATEMENT

We have taken steps to ensure that all experiments in this work are reproducible. Regarding datasets and runtime environments, Section 4 provides the hardware used in our experiments, and all workloads and datasets referenced in the main text are publicly available. We will open source CCKS framework, including the scripts to reproduce the paper results, upon the acceptance of this paper.

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

## A  APPENDIX

### A.1  USE OF LARGE LANGUAGE MODELS (LLMS)

We used LLM-based tools solely for polishing the writing, including improving grammar, phrasing, and readability. No part of the technical content, analysis, or experimental results was generated by an LLM. The authors remain fully responsible for the accuracy and integrity of the paper.

