# OpenReview forum: "CCKS: Cooperative CPU-GPU Scheduling for Fused Kernels on Coherent Architectures"
_ICLR.cc/2026/Conference — Submitted to ICLR 2026_

### Official Review · Reviewer_S4m5 · 2025-11-01

**Soundness:** 3
**Presentation:** 3
**Contribution:** 3
**Rating:** 6
**Confidence:** 4

**Summary:**

This paper presents CCKS, a framework for fused kernel scheduling on tightly coupled CPU/GPU systems, like Grace Hopper.

**Strengths:**

+ Impressive engineering, and great results on a challenging problem to improve performance on shared-memory systems.

**Weaknesses:**

- Can this technique generalize beyond Nvidia Grace Hopper?
- How much of the benefit is truly due to CCKS, versus the underlying shared-memory paradigm of Grace Hopper?

**Questions:**

At a high level, I like this idea, and I believe the authors did a solid job explaining the challenges and presenting their solution. My main question is, "Isn't something like this supposed to already exist on Grace Hopper?" The fact that it doesn't (if it doesn't) highlights that this is important work. However, under the hood, I'm struggling to understand if the speed up is indeed due to CCKS's improved scheduler, or the fact that Grace Hopper is the first, real, shared memory system between CPU/GPU? By virtue of having such direct shared memory, how much benefit is CCKS providing to the end user, versus the CUDA compiler eventually enabling CCKS's ideas?

---

> ### Author Response · Authors · 2025-11-16
>
> Thank you for submitting your review. We plan to address your questions in the revision plan (posted in the general comments) and will post the revised paper on 11/20. Please let us know if you have further important concerns beyond the plan! We will also address your other concerns here if they are not covered in the general reply.

---

> > ### Author Response · Authors · 2025-11-22
> >
> > ## Q: Can this technique generalize beyond Nvidia Grace Hopper?
> >
> > A: Yes. Our technique is not tied to Grace Hopper specifically; instead, it targets a broader architectural trend. Moreover, tightly integrated, cache-coherent CPU–GPU systems are becoming a central part of next-generation hardware roadmaps. For example, upcoming platforms such as NVIDIA GB200 (Grace–Blackwell), the NVIDIA Rubin architecture, and Intel’s Meteor Lake all feature increasingly coherent, high-bandwidth CPU–GPU interconnects. Our work should be viewed as the first step toward exploring new system capabilities enabled by this class of architectures. As these platforms become more widely available, we expect CCKS to apply even more broadly and enable additional use cases built on fast, coherent cross-processor communication.
> >
> > ## Q: How much of the benefit is truly due to CCKS, versus the underlying shared-memory paradigm of Grace Hopper?
> >
> > A: We agree that Grace Hopper exposes a powerful shared-memory abstraction, but no prior work has used these coherence features to build a scheduling mechanism. To our knowledge, CCKS is the first system to exploit GH’s coherent CPU–GPU memory model for fine-grained scheduling.
> >
> > We have added Figure 14 to clarify this point: CCKS leverages **coherence** through speculative enqueueing, load-aware scheduling (Figure 12, CCKS-Aware), and other optimized fast-path mechanisms. These choices directly enable performance improvements beyond what the hardware alone provides. For example, although a GPU-resident scheduler achieves ~0.47 µs SM-to-SM latency (about 8× lower than a CPU-only design), CCKS still provides 77% higher throughput than the GPU scheduler and **28× speedup over a naïve CPU scheduler, even though the interconnect bandwidth advantage is only around 5×**. This shows that CCKS’s benefits arise not just from GH’s shared-memory model, but from algorithmically exploiting coherence to reduce critical-path stalls and scheduling overhead.

---

### Official Review · Reviewer_iDsr · 2025-11-01

**Soundness:** 2
**Presentation:** 2
**Contribution:** 2
**Rating:** 4
**Confidence:** 4

**Summary:**

This paper leverages the cache coherence in modern CPU–GPU systems such as the NVIDIA Grace Hopper Superchip to develop a cache-coherent kernel scheduling framework (CCKS). The framework introduces three techniques: speculative enqueue, batch commit, and CPU bypass, to improve scheduling efficiency. CCKS is integrated into two existing LLM inference systems, Pod-Attention and Mirage.

**Strengths:**

The motivation is clearly articulated and grounded in practical limitations of existing on-GPU scheduling.

The paper is generally well written and easy to follow.

The proposed optimisation techniques are simple yet effective. For instance, in speculative enqueue, when the speculation is incorrect, the committed task can be simply overwritten by the correct task.

**Weaknesses:**

- Grace Hopper is no longer NVIDIA’s latest GPU architecture. It would be good to discuss whether CCKS remains applicable to newer generations such as Blackwell, and what hardware assumptions (e.g., cache coherence model or interconnect behaviour) are required.

- The integration details of CCKS within Pod-Attention and Mirage are limited. More implementation specifics on how speculative enqueue, batch commit, and CPU bypass are realized within these frameworks would improve clarity.

- The paper does not explicitly discuss characteristics of ML workloads that make them particularly suited to CCKS. The current design appears general to GPU workloads. Please elaborate on why ML inference workloads especially benefit from the proposed mechanisms, or what properties (e.g., kernel granularity, dependency patterns) motivate this focus.

- From a venue-fit perspective, this paper is primarily a systems contribution aimed at improving runtime efficiency rather than a study of representation learning. Its relevance to ICLR may therefore be a question.

**Questions:**

The paper states that “traditional inference systems for LLMs execute a sequence of discrete GPU kernels to perform computation.” Could the authors provide a more concrete example illustrating how LLM inference generates a sequence of small kernels, which makes the associated launch and teardown overheads become non-negligible?

When mentioning that “each kernel launch incurs a setup and teardown cost,” it may strengthen the presentation to explicitly link the setup phase to the enqueueing process on the GPU command queue, which is one of the key motivations for the proposed speculative enqueue mechanism.

In speculative enqueue, when a speculation is incorrect, the committed task is overwritten by the correct one. Are there any negative side effects (e.g., wasted memory traffic, resource contention, or timing delays) from frequent mis-speculations? A discussion or quantitative measurement would be helpful.

---

> ### Author Response · Authors · 2025-11-16
>
> Thank you for submitting your review. We plan to address your questions in the revision plan (posted in the general comments) and will post the revised paper on 11/20. Please let us know if you have further important concerns beyond the plan! We will also address your other concerns here if they are not covered in the general reply.

---

> > ### Author Response · Authors · 2025-11-22
> >
> > Q: **Grace Hopper is no longer NVIDIA’s latest GPU architecture. It would be good to discuss whether CCKS remains applicable to newer generations such as Blackwell, and what hardware assumptions (e.g., cache coherence model or interconnect behaviour) are required.**
> >
> > A: We are currently working to obtain access to GB200 hardware and will update our evaluation once it becomes available. GB200 continues to provide the same coherent CPU–GPU interconnect as GH200, maintaining the hardware properties that CCKS relies on. Given these similarities, we expect CCKS to deliver comparable performance benefits on GB200.
> >
> > ---
> >
> > Q: **The integration details of CCKS within Pod-Attention and Mirage are limited. More implementation specifics on how speculative enqueue, batch commit, and CPU bypass are realized within these frameworks would improve clarity.**
> >
> > A: We added integration details in the revised §4.1. Importantly, these optimizations (e.g., *Speculative Enqueue*, *Batch Commit*, and *Scheduler Bypass*) are implemented inside the CCKS library itself. Thus, applications automatically benefit from these mechanisms when using the CCKS API, without any additional code modifications.
> > For example, we implement the Mirage scheduler following the computation graph generated by Mirage, and apply speculative enqueue to pre-populate tasks for the next token instead of waiting for the previous token to finish.
> >
> > ---
> >
> > Q: **The paper states that “traditional inference systems for LLMs execute a sequence of discrete GPU kernels to perform computation.” Could the authors provide a more concrete example illustrating how LLM inference generates a sequence of small kernels, which makes the associated launch and teardown overheads become non-negligible?**
> >
> > A: Thank you for this detailed comment! We revised §2.1 to better articulate why dynamic persistent kernels are essential. LLM inference requires:  *adaptability to dynamic runtime inputs* (e.g., batch size, sequence length).
> > These demands are uniquely well served by dynamic persistent kernels and are difficult to satisfy using traditional GPU execution models.
> >
> > We also include concrete evidence illustrating the scale of kernel launch overhead in LLM inference:
> > - **KTransformer (SOSP’25)** reports Fiddler (ICLR 25) has *7,000 kernel launches per decoded token*, each averaging 16 µs, with **73%** of execution time spent purely on launch overhead.
> > - **Llama.cpp** reduces this to ~3,000 launches per token via CUDA Graphs and aggressive fusion, yet launch overhead still accounts for **21%** of total GPU time.
> > - **FlashMoE (NeurIPS’25)** reports **300+** kernel launches for MoE communication alone.
> >
> > As newer LLM architectures increase parallelism and modularity, this overhead trend becomes even more pronounced.
> >
> > ---
> >
> > Q: **When mentioning that “each kernel launch incurs a setup and teardown cost,” it may strengthen the presentation to explicitly link the setup phase to the enqueueing process on the GPU command queue, which is one of the key motivations for the proposed speculative enqueue mechanism.**
> >
> > A: Dynamic persistent kernels partially mitigate setup and teardown costs by overlapping these phases with other GPU work. However, realizing this benefit in practice still requires careful *scheduling*, particularly when workloads have dynamic dependencies. CCKS directly targets this scheduling problem: it ensures these intended overlaps actually materialize and are not negated by dependency resolution, host-side scheduling delays, or suboptimal scheduling choices.
> >
> > For example, **ThunderKittens** provides kernel templates (Paper §Programming Abstractions) that split execution into multiple phases, enabling explicit runtime overlap. CCKS generalizes and automates these benefits by providing a CPU-driven speculative scheduler that ensures enqueue–execution overlap even under workload dynamism.
> >
> > ---
> >
> > Q: **In speculative enqueue, when a speculation is incorrect, the committed task is overwritten by the correct one. Are there any negative side effects (e.g., wasted memory traffic, resource contention, or timing delays) from frequent mis-speculations? A discussion or quantitative measurement would be helpful.**
> >
> > A: We added Figure 13 to analyze the main source of nondeterminism in CCKS—speculative enqueueing. Across our evaluated workloads, mis-speculations not always introudce extra overhead (for example, changing the tasks that to be executed for load balancing). Overall, it introduce only about **~1% performance overhead**, with negligible effects from additional memory traffic or contention. At the same time, the fast-response CPU scheduler in CCKS enables substantial latency-critical benefits, including:
> > - SM-workload–aware load balancing (Figure 12 “CCKS-Aware”),
> > - dynamic computation graph support, and
> > - new scheduling policies enabled by CPU-side reasoning.
> >
> > These benefits significantly outweigh the small overhead of occasional mis-speculations.

---

### Official Review · Reviewer_D16m · 2025-11-02

**Soundness:** 3
**Presentation:** 3
**Contribution:** 3
**Rating:** 6
**Confidence:** 4

**Summary:**

The paper proposes a new scheduling framework, CCKS, for fused persistent GPU kernels that leverages emerging cache-coherent CPU–GPU architectures such as NVIDIA’s Grace Hopper with low latency high bandwidth links. Recent fused kernel systems execute scheduling logic on the GPU that cause inefficiency due to control-heavy, serial scheduling running on SIMT hardware. CCKS instead offloads this scheduling to the CPU, with optimizations such as speculations and scheduling bypass to make this more efficient.

**Strengths:**

* Tackles an important and challenging problem
* An interesting un-intuitive approach to offload scheduling to the CPU
* Reasonable approach that uses speculation and CPU bypass when needed
* The paper is well motivated
* Significant speedups over baseline approaches

**Weaknesses:**

* The approach relies on a ultra-low latency, high-bandwidth, cache-coherent interconnect between the CPU and GPU. It would be good to see what the impact of latency is and when this approach becomes feasible.
* While the proposed approach is quite interesting, it does add a lot of scheduling complexity and non-determinism in performance to the scheduling pipeline
* Somewhat narrow in applicability, as this would be only useful when using fused persistent kernels.
* Some prior works missing, e.g., "ACE: Efficient GPU Kernel Concurrency for Input-Dependent Irregular Computational Graphs", Durvasula et al., PACT 2024
* There are many typos and grammatical errors in the paper. Please fix them.

**Questions:**

Please comment on/address weaknesses above.

---

> ### Author Response · Authors · 2025-11-16
>
> Thank you for submitting your review. We plan to address your questions in the revision plan (posted in the general comments) and will post the revised paper on 11/20. Please let us know if you have further important concerns beyond the plan! We will also address your other concerns here if they are not covered in the general reply.

---

> ### Author Response · Authors · 2025-11-22
>
> ## Q1. The approach relies on an ultra-low-latency, high-bandwidth, cache-coherent interconnect between the CPU and GPU. It would be good to see what the impact of latency is and when this approach becomes feasible.
>
> **A:** In the revised paper, we added **Figure 14**, which reports both **operation latency** and **end-to-end latency** when implementing CCKS over PCIe. Our evaluation shows that PCIe introduces up to a **28× slowdown** compared to a coherent interconnect—significantly slower than even discrete kernel launches.
>
> Importantly, CCKS’s benefits come not only from lower latency but also from **specialized coherent features** (e.g., fine-grained synchronization and load/store access patterns) that PCIe cannot provide.
>
> We also highlight that tightly integrated, coherent CPU–GPU systems are already appearing on hardware roadmaps. Examples include **NVIDIA GB200 (Grace Blackwell)**, the **NVIDIA Rubin platform**, and **Intel Meteor Lake**. Our work serves as an early exploration of compelling use cases enabled by these architectures.
>
> ---
>
> ## Q2. While the proposed approach is interesting, it adds scheduling complexity and performance non-determinism to the scheduling pipeline.
>
> **A:** We added **Figure 13** to illustrate the primary source of nondeterminism—**speculative enqueuing**. Across our evaluated workloads, mispredictions introduce only about **1% performance overhead**, demonstrating that nondeterminism has minimal impact.
>
> At the same time, our results shows CCKS enables several **latency-critical scheduling benefits**, including:
> - **SM-workload-aware load balancing** (Figure 12, *CCKS-Aware*)
> - **Dynamic computation graph updates**  (Figure 13)
> - **New scheduling policies** that are impractical with traditional CPU–GPU synchronization  (Figure 12, *CCKS-Dynamic*)
>
> These results show that the added complexity is small compared to the performance and flexibility that CCKS unlocks.
>
> ---
>
> ## Q3. The approach appears somewhat narrow in applicability, since it is only useful with fused persistent kernels.
>
> **A:** Our target class—**dynamic persistent kernels**—is rapidly gaining traction. For example, **NVIDIA has incorporated dynamic persistent kernel launch with Blackwell Cluster Launch Control in CUTLASS libaray** for its latest Blackwell GPUs.
>
> Furthermore, **CCKS’s optimizations are reusable across different DPK implementations**, making it broadly applicable to systems adopting this increasingly common execution model.
>
> In the revision, we also expanded our discussion in the overview section, including:
> - Existing approaches to reducing kernel-launch overhead
> - Why kernel-launch reduction alone is insufficient for many modern workloads
> - Additional justification for CCKS’s broader relevance
>
> We also revised the paper fixing the format and typo issues. Please let us know if you have any further questions!

---

### Official Review · Reviewer_yXoY · 2025-11-14

**Soundness:** 1
**Presentation:** 2
**Contribution:** 1
**Rating:** 0
**Confidence:** 4

**Summary:**

This paper proposes a framework for scheduling work on the GPU from the CPU exploiting the tight coupling on NVIDIA's Hopper for this purpose.

**Strengths:**

The paper is relatively easy to read.

**Weaknesses:**

The paper appears to me to not accurately describe the key related works it sets out to build upon.  To me it appears neither Wu et al., 2025 or Spector et al., 2024 employ on GPU scheduling (which makes little sense).  My understanding is the scheduling in those papers done offline by the compiler when generating the kernels before they are run.  Also, I didn't see mention of persistent kernels in the OSDI paper from Wu et al., 2025.   Similarly, the statement "The core innovation enabling kernel fusion is an on-GPU scheduler" seems inaccurate or needs some clarification.  Kernel fusion can be done statically before running the code.

The optimizations in Figure 4 and 5 are not explained in nearly sufficient detail to understand where the supposed benefits are coming from.   Partly, that can be blamed on the format of ICLR which has a very limited page budget.  More details could have been provided in an appendix.  Or better, yet, submit to a systems conference where you get twice as much space in the main text.  At a systems conference I would expect a more thorough explanation (with data) of the source of the problem being tackled.

Line 303: "Speculative enqueue: The CPU scheduler speculatively prepares task data structures and copies
them into GPU memory queues in advance, even when prior GPU tasks are still running."  -- GPUs already do this kind of thing since the first CUDA enable GPUs.   The whole point of async memcpy and streams is to allow the CPU to load up work into a ring buffer queue of tasks that are read in by the GPU as the GPU completes work.  GPUs work this way for graphics as well (not just compute).

I don't see mention of CUDA graphs, which seems related.

Typos: "imporve fused kernel", "tighyly integrated"

**Questions:**

Is code available?  I didn't see any supplemental materials or links.

This is a systems paper, which seems a bit outside the normal scope for ICLR.  To my judgment this paper would likely get rejected at flagship systems conferences that are all interested in work on ML, so why should it be published at ICLR?

Where in Wu et al. (2025) [Mirage paper at OSDI 2025] is there a description that matches the following text from this submission "This model designates one or more SMs to act exclusively as schedulers Wu et al. (2025)."?  The words "designate" and "scheduler" do not seem to appear in the OSDI 2025 paper.

---

> ### Author Response · Authors · 2025-11-16
>
> Thank you for your review. We will upload our revised paper by 11/20, which will incorporate all improvements detailed in our general revision plan.
>
> We would also like to offer a few clarifications on specific points in the review:
>
> **Regarding the Paper's Scope**: The reviewer commented: 'This is a systems paper, which seems a bit outside the normal scope for ICLR'. We submitted to ICLR based on the official Call for Papers, which explicitly lists "Infrastructure, software libraries, hardware" as in-scope topics. We believe our work is a strong fit for this area and hope it will be of interest to the ICLR community.
>
> **Regarding the Persistent Kernel Scheduler citations**: The reviewer claims: "didn’t see mention of persistent kernel scheduler in the cited paper" and questions if the referenced work (Mirage) is related to persistent kernel schedulerThis is factually incorrect:
> In Section 2.2, where we introduce persistent kernels schedulers, we explicitly cite Mirage’s we(Jia, 2025, https://zhihaojia.medium.com/compiling-llms-into-a-megakernel-a-path-to-low-latency-inference-cf7840913c17 ). This reference is all about Mirage’s persistent kernel design and has a scheduler section detailing the implementation of the scheduler. We also cite code repository (Mirage-project, 2025: https://github.com/mirage-project/mirage), which contains extensive documentation on the Mirage’s persistent kernel and number of SMs designated to scheduler.  These reference appears multiple time in the paper and is hard to miss.
>
> We respectfully request the AC consider whether this review meets ICLR's standards for professional, informed peer review.

---

> ### Author Response · Authors · 2025-11-22
>
> We have updated the paper to clarify the key concerns raised. Below, we address specific points regarding related work, citations, scope, and open-source code.
>
> 1. **Paper Out of Scope**: We submitted to ICLR based on the official Call for Papers, which explicitly lists "Infrastructure, software libraries, hardware" as in-scope topics. We believe our work is a strong fit for this area and hope it will be of interest to the ICLR community. Given the growing importance of system impact on LLM performance (see the DeepSeek-V3 Technical Report), we believe our contributions to LLM inference optimization are highly relevant. Empirically, our system demonstrates significant practical value, achieving up to a 77% speedup over state-of-the-art baselines on representative workloads.
>
> 2. **Mirage Citation and Persistent Kernels** The reviewer asked where Mirage discusses persistent kernels. We cited (multiple times) in our submission on Mirage’s most recent support for persistent kernels, which is a development introduced after their original OSDI paper.
>   - Clarification: In our original submission (Section 2.2), when we first introduced the persistent kernel, we referenced Mirage’s recent technical post (Jia, 2025) and their code repository multiple times, which details the design of their persistent kernel and scheduler in Mirage.
>   - Revision: To avoid confusion with the older OSDI publication, we have updated the reference to ensure that, wherever we cite Mirage, we point exclusively to the most recent 2025 technical blog and the official GitHub repository, which contain the relevant scheduler implementation details.
>
> 3. **Clarification on Related Works and Background** We have revised the background discussion in Section 2 to contextualize our contributions better. Specifically, in the subsection titled “From Discrete Kernels to Dynamic Persistent Kernels,” we now explicitly contrast the limitations of traditional discrete kernels and CUDA Graphs with the capabilities of recent persistent kernel architectures. We also noticed CUDA graph cannot fully resolve the kernel launch overhead. For example,  KTransformer (SOSP25) claims LLama.cpp with low level optimizations (including enabling CUDA Graph) and aggressive kernel fusion, the kernel launch overhead still represents 21% of total GPU execution time.
>
> 4. **Code Availability and Open Source** We are fully committed to reproducibility. We will provide the core implementation code in the supplementary material. Upon acceptance, we will release the full source code, including the integration examples for Mirage, Pod-Attention, and ThunderKitten, on GitHub under an open-source license.
>
>  Please also let us know what performance details you would like to learn from.

---

### Author Response · Authors · 2025-11-16

**[Update: 11/22: The revised version is uploaded] **

We appreciate the reviewers’ feedback. Below, we describe our plan for addressing review’s common concerns (**weaknesses:W**) and thoughtful questions (**questions:Q**).

Writing and Clarity Improvements

- **Background and Motivations [R1, R2, R3, R4]**: We will include more details about how the megakernel compares to static approaches like CUDA Graph [R1], how it is generated, and how small tasks are launched. This will include experimental data highlighting the overhead breakdown (e.g., setup and launch overhead) in existing systems. [R3: Q1, Q2], we will also add more citations, including ACE [R2: W2], and prior research on CPU-side scheduling [R4: Q1].
- **More integration details [R3]**: We will clarify integration details, including how CCKS fits into existing command queuing systems and how it compares to existing systems' scheduling capabilities. [R3: W2]
- **We will fix all typos [R1,R2]**.

New Experiments [R2, R3, R4]

- **Interconnect Performance [R2, R4]**: We will include a breakdown highlighting the impact of cache coherence. [R2: W1, R4: Q2], and will experiment on CCKS under different CPU/GPU interconnect latencies.
- **Newest GPU Architecture [R3]**: We will report CCKS results on the latest B200 systems. [R3: Q1]
- **Speculative enqueue [R3]**: We will provide an analysis of speculative enqueue, including the number of misses in our execution and the total overhead it causes. [R3: Q3]

Open-source code [R1]

We will include our open-source, working code in the supplementary material of the revised paper.

(R1: by Reviewer yXoY, R2: by Reviewer D16m, R3: by Reviewer iDsr, R4: by Reviewer S4m5)

---

### Meta-Review · Area_Chair_W4Mz · 2026-01-11

**Summary:**

I recommend rejection for this submission, though I would like to acknowledge the solid engineering effort behind CCKS. My primary concern is that it remains unclear how much of the performance improvement stems from the proposed scheduling algorithms versus the inherent advantages of Grace Hopper's cache-coherent shared-memory architecture. While the authors argue that the 77% throughput gains exceed what interconnect bandwidth alone could explain, I find this argument not fully convincing without broader hardware validation. The missing Blackwell evaluation, which was promised during rebuttal, leaves an important gap in demonstrating that CCKS generalizes beyond the specific Grace Hopper platform.

**Reviewer Concerns:**

The authors put considerable effort into addressing reviewer feedback with new experiments and clarifications. One reviewer's strong rejection was based on factually incorrect claims about missing citations, so that assessment carries reduced weight. That said, key concerns persist. Reviewers questioned whether gains come from CCKS's algorithms or simply from Grace Hopper's shared-memory hardware. The authors argue that 77% throughput improvements exceed the ~5× interconnect bandwidth advantage (Figure 14) and that speculative mispredictions add only ~1% overhead (Figure 13). However, promised Blackwell results are missing, leaving generalizability to newer architectures undemonstrated.

**Reviewer Scores:**

Discounting the problematic review, the remaining three scores (4, 6, 6) still reflect a borderline paper. Two reviewers placed it marginally above the threshold, recognizing solid engineering work. However, despite the authors' efforts to address concerns about interconnect performance and scheduling overhead, the responses did not sufficiently resolve questions about generalizability and the missing Blackwell evaluation remains a gap.

---

### Decision · Program_Chairs · 2026-01-26

Reject